# Nonmalignant Features Associated with Inherited Colorectal Cancer Syndromes-Clues for Diagnosis

**DOI:** 10.3390/cancers14030628

**Published:** 2022-01-26

**Authors:** Diana Haimov, Sari Lieberman, Sergi Castellvi-Bel, Maartje Nielsen, Yael Goldberg

**Affiliations:** 1Raphael Recanati Genetic Institute, Rabin Medical Center, Beilinson Hospital, Petach-Tikva 4941492, Israel; dianaha2@clalit.org.il; 2Medical Genetics Institute, Shaare Zedek Medical Center, Jerusalem 9103102, Israel; sari@szmc.org.il; 3Department of Gastroenterology, Institut d’Investigacions Biomèdiques August Pi i Sunyer (IDIBAPS), Centro de Investigación Biomédica en Red de Enfermedades Hepáticas y Digestivas (CIBERehd), Hospital Clínic, 08035 Barcelona, Spain; SBEL@clinic.cat; 4Department of Clinical Genetics, Leiden University Medical Center, Albinusdreef 2, 2333 ZA Leiden, The Netherlands; M.Nielsen@lumc.nl; 5Sackler Faculty of Medicine, Tel Aviv University, Tel Aviv 6997801, Israel

**Keywords:** colorectal cancer, genetic predisposition, diagnosis, extracolonic, germline, mosaicism, mTOR polyposis, *TGF-β*, WNT

## Abstract

**Simple Summary:**

Familiarity with nonmalignant features and comorbidities of cancer predisposition syndromes may raise awareness and assist clinicians in the diagnosis and interpretation of molecular test results. Genetic predisposition to colorectal cancer (CRC) should be suspected mainly in young patients, in patients with significant family histories, multiple polyps, mismatch repair-deficient tumors, and in association with malignant or nonmalignant comorbidities. The aim of this review is to describe the main nonmalignant comorbidities associated with selected CRC predisposition syndromes that may serve as valuable diagnostic clues for clinicians and genetic professionals.

**Abstract:**

Genetic diagnosis of affected individuals and predictive testing of their at-risk relatives, combined with intensive cancer surveillance, has an enormous cancer-preventive potential in these families. A lack of awareness may be part of the reason why the underlying germline cause remains unexplained in a large proportion of patients with CRC. Various extracolonic features, mainly dermatologic, ophthalmic, dental, endocrine, vascular, and reproductive manifestations occur in many of the cancer predisposition syndromes associated with CRC and polyposis. Some are mediated via the WNT, *TGF-β*, or mTOR pathways. However the pathogenesis of most features is still obscure. Here we review the extracolonic features of the main syndromes, the existing information regarding their prevalence, and the pathways involved in their pathogenesis. This knowledge could be useful for care managers from different professional disciplines, and used to raise awareness, enable diagnosis, and assist in the process of genetic testing and interpretation.

## 1. Introduction

The molecular diagnostics of hereditary colorectal cancer (CRC) has advanced significantly in recent decades, enabling the development of efficient screening strategies to identify individuals at high risk for referral for genetic testing and specialized cancer surveillance programs. Genetic diagnosis of affected individuals and predictive testing of their at-risk relatives, combined with intensive cancer surveillance, has an enormous cancer-preventive potential in these families. At the same time, it reduces anxiety among family members found to be mutation-negative and spares them from undergoing unnecessary procedures. 

The majority of CRCs occur sporadically. The inherited germline contribution stands at between 12% and 35% of all cancer cases [1,2], and only 5–7% of CRC cases are attributed to germline mutations in genes that are responsible for Mendelian cancer syndromes [3]. About 10% of individuals with CRC are less than 50 years old; in 40% of them CRC is due to a genetic condition [4,5].

Genetic predisposition to CRC should be suspected in patients who are young, have multiple polyps, or have mismatch repair-deficient (MMRd) tumors, especially at a young age or in association with malignant or nonmalignant comorbidities. Sometimes, the clinical features are mild and the patient does not meet the diagnostic criteria. One of the reasons for milder phenotypes is somatic mosaicism. Genetic variants that arise post-zygotically are not present in all body cells, posing a unique diagnostic challenge. Mosaicism in CRC susceptibility genes, mainly *APC*, has been reported in a significant portion of patients with undiagnosed polyposis [6,7,8]. In addition to the milder clinical phenotype, which may be interpreted as a “normal variation”, cases of mosaicism are likely to be underdiagnosed because of the limited sensitivity of conventional molecular diagnostic techniques, insufficient material used for testing, and the false assumption that when a family history is absent the patient’s cancer is sporadic [6]. Accordingly, mosaic variants that result in a milder phenotype in probands may have limited expression, occurring mainly in extra-colonic tissues. For example, the literature contains descriptions of a carrier of a mosaic alteration of *PTEN* in whom the phenotype was restricted to macrocephaly and Hashimoto’s thyroiditis [9] and carriers of a mosaic alteration of *APC* presenting with desmoid tumors without or prior to the appearance of colonic polyps [10]. 

Familiarity with such nonmalignant features/comorbidities in CRC and polyposis predisposition syndromes may assist clinicians in the diagnosis and interpretation of molecular results in several ways (Table 1). First, it may raise clinical suspicion and prompt the referral of patients for genetic testing that will confirm the presence of cancer or polyposis or identify healthy genetic carriers with nonmalignant features, leading to cancer prevention. Second, it may guide the clinician to select the most appropriate test, i.e., a limited or extended gene panel, or other tests such as chromosomal microarray. Third, it may facilitate the interpretation of variants when results are inconclusive, for example, variants of unknown significance (VUS) or low-level mosaicism. A lack of awareness may be part of the reason why the underlying germline cause remains unexplained in a large proportion of patients with CRC.

The aim of this review was to describe the main nonmalignant comorbidities associated with CRC predisposition syndromes that may serve as valuable diagnostic clues for clinicians and genetic professionals.

## 2. Inherited Colorectal Cancer Predisposition Syndromes

Mendelian hereditary CRC syndromes are mainly due to germline mutations in *APC, MUTYH*, and the MMR genes (*MSH2, MSH6, PMS2*, and *MLH1*), which cause, respectively, familial adenomatous polyposis (FAP; #175100); *MUTYH-*associated polyposis (MAP; #608456); Lynch syndrome )#120435); or constitutional mismatch repair deficiency (CMMRD; MIM #276300) in the biallelic mode [3,11]. Other inherited CRC genes include *BMPR1A* and *SMAD4*, which lead to juvenile polyposis (#174900); *PTEN*, which leads to *PTEN* hamartoma tumor syndrome (PHTS; (#158350); *AXIN2*, which leads to oligodontia-colorectal cancer syndrome (#608615); *STK11-*Associated Peutz–Jeghers Syndrome *(#175200), GREM1*, which leads to hereditary mixed polyposis syndrome (HMPS; (#601228); and *POLD1* and *POLE*, which lead to polymerase proofreading-associated polyposis (PPAP; (#615083). Also linked to CRC are *BLM*; *MCM8* and *MCM9*; *NTHL1*; and *RNF43* (Table 2) [12,13,14,15,16,17,18]. Many of the predisposition syndromes have nonmalignant features. In this review, we elaborate on the most relevant features that contribute to diagnosis (Table 2).

### 2.1. APC-Associated Familial Polyposis (FAP)

FAP is a dominantly inherited disease caused by germline mutations in the *APC* gene [19]. FAP carriers are at very high risk of colorectal cancer, and at increased risk of gastric, small bowel, pancreas, and thyroid carcinoma, as well as medulloblastoma and pediatric hepatoblastoma. Benign extraintestinal manifestations develop in the majority (70%) of patients, including desmoid tumors (10–25% of patients); congenital hypertrophy of the retinal pigmented epithelium (CHRPE; over 90% of patients); dental abnormalities (oligodontia among others); fundic gland polyps; Gardner fibromas; osteomas; adrenal masses (mostly adrenocortical adenomas); and epidermoid cysts or lipomas on any part of the body [19,20]. Attenuated FAP (AFAP) is diagnosed if less than 100 adenomas are observed in the colon. However, the number of extra-intestinal manifestations seems to be lower in AFAP [21]. Historically, the combination of colonic polyposis and prominent extra-intestinal manifestations was termed Gardner’s syndrome after Eldin Gardner, who originally described the association [22]. 

Since 15–20% of cases of FAP occur de novo, a portion with somatic mosaicism, some of these manifestations may represent a sentinel sign of FAP preceding the gastrointestinal symptoms and may help in establishing an early diagnosis. Early referral of patients for colonic screening is important to prevent serious complications.

Desmoid tumors may appear at a younger age in familial FAP than they do in sporadic FAP, mostly in the abdomen and small bowel mesentery, particularly after surgery and pregnancy. They can also arise in the shoulder girdle, chest wall, or inguinal regions [23,24]. About 11–15% arise due to germline *APC* mutations; one study of 12 affected patients found that at least 4 carried a germline *APC* variant (33%); most (65%) are due to somatic *CTNNB1* mutations [24,25,26]. Since germline *APC* variants and somatic *CTNNB1* mutations are mutually exclusive, *CTNNB1* molecular testing has been suggested as the first diagnostic step [26]. 

One study reported a germline *APC* mutation in three out of seven patients (43%) with apparently sporadic Gardner fibroma [27]. There are also single case reports of Gardner fibroma presenting before age 1–2 years with germline *APC* mutations [19,28].

The literature to date contains no large studies of the prevalence of *APC* germline mutations in patients with osteomas. However, there are reports of a child with osteoma and no colorectal phenotype at the time of diagnosis who was found to have a de novo frameshift *APC* mutation [29], and of a 16-year-old patient with Gardner fibroma as well as osteoma [30] with a germline *APC* mutation. A subsequent colonoscopy in the latter case revealed more than 100 small sessile polyps.

Additionally, germline *APC* mutations were reported in two studies in 10% [31] and 14% [32] of 29 patients with apparently sporadic hepatoblastoma. However, a third study, with 50 patients, found that germline *APC* mutations were rare in this patient group [33]. Similar to desmoid tumors and hepatoblastoma, somatic *CTNNB1* mutations are common in sporadic hepatoblastoma (50–90% of patients) and likely mutually exclusive with germline *APC* mutations, although further research is needed [32].

CHRPE has been reported in 1.2–4.4% of individuals from the general population who underwent fundus examination [34]. Thus, even if it occurs in infancy, CHRPE as a sole feature is not enough to raise suspicions and prompt genetic testing. However, unlike sporadic CHRPE, hereditary CHRPE has a tendency for multiplicity and bilaterality [35]. It is possible that young patients with bilateral CHRPE are more likely to have a germline *APC* mutation, but this needs to be further studied. A correlation between the site of the mutations and expression of CHRPE has been reported [29], with CHRPE being reported mostly in patients with mutations in 3’ to exon 9 and not beyond codon 1444. However, 18% of patients with point mutations did not conform to this rule [29]. It should be emphasized that FAP-related CHRPE may not be seen in all patients with FAP.

A specific group of patients with polyposis and mental retardation may present with large (usually de novo) chromosomal deletions encompassing the 5q22 region which harbors the *APC* gene. Indeed, mental retardation and dysmorphic features are usually the reason for genetic testing in these patients, with polyposis or CRC diagnosed only at a later stage. Currently, about 20 cases have been reported in the literature. In only 4 was the exact deletion site identified by chromosomal microarray (CMA) testing [36,37]. According to one group, deletions encompassing the distal portion of the region, from 5q22 to 5q31.1, may lead to a more severe phenotype with severe mental retardation and hypotonia, whereas the absence of the proximal portion, from 5q15 to 5q22.2, is associated with mild mental retardation, minor dysmorphic features, and possible organ abnormalities such as horseshoe kidneys [38].

It is likely that the many other genes located in the deleted regions also play a role in these predisposition manifestations and mental retardation [39]. A patient with a 5q22.2q23.1 deletion with FAP but without mental retardation or dysmorphic features was reported [37]. This finding was attributed to the small size of the deleted region relative to previously reported cases, harboring fewer genes. Notably, the deleted region contained *TSSK1B*, and the absence of this gene probably explained the asthenozoospermia diagnosed in this patient. 

Based on findings that the *APC* gene plays a role in the development of the central nervous system [40], and that conditional *APC* knockout mice show learning and memory impairments and autistic-like behaviors [41], researchers have hypothesized that mutations in the *APC* gene itself may lead to developmental delay/mildly impaired intellectual abilities. However, the available data on the cognitive abilities of patients with truncating *APC* mutations are sparse and somewhat conflicting. In 2020, Cruz-Correa et al. [42] compared cognitive ability among 26 individuals with FAP (mean age 34 ± 15.0 years) and 25 age-, gender-, and educational-level-matched controls (mean age 32.7 ± 13.8 years). The patients with FAP had a significantly lower IQ (*p* = 0.005) and performed significantly worse on all tasks of the Batería III Woodcock–Muñoz test. They also scored within the deficient range on long-term retrieval and cognitive fluency. In 2016, Azofra et al. [43] used a sibling-pair design to compare three patients with FAP with their sex-matched siblings without FAP. Siblings in all three pairs had similar general intelligence, executive function, and basic academic skills, with no differences in brain structure on magnetic resonance imaging. However, there were subjective differences in behavioral and emotional characteristics as assessed by maternal perception.

### 2.2. AXIN2-Associated Oligodontia–Colorectal Cancer Syndrome

Heterozygosity for germline variants in *AXIN2* and its relation to tooth anomalies, gastrointestinal polyps, and the risk of CRC was first described in 2004 [44]. One group diagnosed up to 100 colon polyps in carriers of *AXIN2* variants from age 31 years [45]. Cases of germline variants in *AXIN2* related to ectodermal dysplasia have been reported as well [46,47].

Several researchers hypothesized that when the AXIN2 protein is mildly affected, as in cases of missense variants, individuals are more prone to non-syndromic oligodontia, whereas truncating pathogenic variants in *AXIN2* are more likely to predispose carriers to both oligodontia and CRC [48,49]. However, *AXIN2* variants previously found in patients with apparently non-syndromic oligodontia may add to the risk of polyposis/CRC later in life. This finding emphasizes the importance of early referral of individuals with *AXIN2*-related tooth agenesis for colonoscopy surveillance. The National Comprehensive Cancer Network (NCCN) suggests that surveillance with colonoscopy for carriers should begin at age 25–30 years.

### 2.3. BMPR1A- and SMAD4-Associated Juvenile Polyposis

Juvenile polyposis syndrome (JPS) is a rare hamartomatous condition with an autosomal dominant pattern of inheritance. Loss of function mutations in *SMAD4* and *BMPR1A* account for the majority of inherited cases. The characteristic multiple benign polyps may cause bleeding and anemia, but their significant consequence is malignant transformation. The risk of polyposis and colon cancer is similar for carriers of *SMADI4* or *BMPR1A* mutations, but *SMAD4* is associated with additional tumoral and other phenotypes. *SMAD4* mutation carriers have a higher prevalence of gastric polyps and extracolonic cancer, such as testicular cancer [50]. *SMAD4*-JPS phenotypes include hereditary hemorrhagic telangiectasia (HHT), a vascular dysplasia affecting 76% of carriers. HHT may manifest in early childhood and can be life-threatening [51]. It is associated with epistaxis (61–71%), telangiectasias (57%), and arteriovenous malformation (pulmonary, visceral, hepatic, or cranial) [52]. A smaller proportion of *SMAD4* mutation carriers may also present with thoracic aortic disease (e.g., aortic root dilatation, aneurysm, and aortic dissection), mitral valve dysfunction, and, rarely, other connective tissue disorders (retinal detachment, brain aneurysm, and lax skin and joints) [53]. 

Copy number variations, including the 10q22–23 deletions, account for 15% of *BMPR1A-*associated genetic alterations, which can include the deletion of *BMPR1A* alone or the contiguous deletion of both *BMPR1A* and *PTEN*. When both genes are absent, the phenotype is more severe in terms of the age of onset and may include symptoms related to mutations in *PTEN* (see below).

### 2.4. MCM8- and MCM9-Associated CRC

*MCM8* and *MCM9* were recently added to the list of genes causing CRC with an autosomal recessive pattern of inheritance [18,54]. Affected patients also presented with primary ovarian failure [54] or reproductive disorders [19] including azoospermia [55]. Although infertility is suspected to have a strong genetic basis, in the majority of cases there is no established cause. Nevertheless, clinicians encountering infertile patients diagnosed with mutated *MCM8* or *MCM9* should consider referral to a gastroenterologist for early surveillance. 

### 2.5. MMR-Associated Polyposis

Lynch syndrome, or hereditary nonpolyposis CRC, is caused by an inherited alteration in one of the MMR genes (*MLH1* (OMIM,120436), *MSH2* (OMIM, 609309), *MSH6* (OMIM, 600678), *PMS2* (OMIM, 600259) and *EPCAM* (OMIM, 185535)). Lynch syndrome is also associated with increased risk of other malignancies, mainly for endometrial, but also for ovarian, renal, and pancreas cancer, as well as glioblastoma. It is usually not associated with benign features. The prevalence in the general population is approximately 0.3% [56]. 

Muir–Torre syndrome (MTS) is an uncommon variant of Lynch syndrome associated with sebaceous neoplasms of the skin, ranging from sebaceous gland hyperplasia to sebaceous adenoma, epithelioma, and sebaceous carcinoma. The neoplasms show a loss of MSH2 expression in over 90% of cases, followed by a loss of MLH1 [57]. A study of 205 patients with MTS showed that in 22%, cutaneous sebaceous neoplasms (at any site) appeared before the visceral malignancy [58]. Several groups attempted to establish screening guidelines for the evaluation of MTS in patients presenting with sebaceous neoplasms [59]. 

CMMRD, the biallelic form of Lynch syndrome, is a rare cancer predisposition syndrome associated with high penetrant cancer in childhood [11]. The tumor spectrum also includes gastrointestinal polyposis; CRC; and mostly lymphomas, leukemias, and brain tumors [60]. Various extracolonic features have been described in children with CMMRD [11]. There is a clear phenotypic overlap between CMMRD and neurofibromatosis type 1 (NF1), with 62–97% of patients with CMMRD presenting with café-au-lait macules (CALMs) and 10% with both CALMs and axillary and/or inguinal freckling. Several anecdotal reports of other *NF1* features have been published. As listed in the C4CMMRD consensus guidelines, there are a number of non-neoplastic features, such as pigmentary alterations, pilomatrichomas, agenesis of the corpus callosum, non-therapy-induced cavernomas, and multiple developmental vascular abnormalities in separate regions of the brain. Their presence should raise suspicions of a potential diagnosis of CMMRD in young cancer patients or patients with suspected *NF1* but without an *NF1* or *SPRED1* pathogenic variant [61]. Children with CMMRD may also have nonmalignant pediatric systemic lupus erythematosus and features typical of tuberous sclerosis [62].

### 2.6. MUTYH-Associated Polyposis

MAP, first described in 2002, is caused by biallelic mutations in *MUTYH* [63]. It is typically associated with ten to a few hundred adenomatous colonic polyps with an increased risk for CRC. Duodenal polyps, osteomas, dental anomalies, and CHRPE have been described in patients with MAP. One patient was reported to have small bowel tumors and intra-abdominal desmoids [64]. Others reported jawbone cysts in 11 of 276 patients with MAP [65]. CHRPE has an estimated prevalence of 5.5% in patients with MAP, although this figure may be overestimated as pigment anomalies of the retina are quite frequent in the general population [65]. At least in one case, authors could not confirm that the CHRPE observed was specifically of the type associated with polyposis [65,66].

### 2.7. NTHL1-Associated Polyposis

Biallelic germline *NTHL1* mutations predispose carriers to the development of *NTHL1*-associated polyposis (NAP) and CRC [15]. Molecular and clinical analysis of 29 carriers of biallelic germline *NTHL1* mutations yielded a broad spectrum of cancers associated with this syndrome, in addition to 14 types of premalignancies and benign tumors, including skin hemangiomas, ovarian and liver cysts, and neurofibromas in a few cases [67].

### 2.8. POLE- and POLD1-Associated Polyposis

Germline mutations affecting the proofreading domains of *POLE* and *POLD1*. (OMIM#615083 and OMIM#612591, respectively), causing *POLE*- and *POLD1*-associated polyposis (PPAP), predispose carriers to the development of colorectal adenomas and carcinomas and endometrial carcinomas, [14]. Nonmalignant features are not considered part of PPAP, although CALMs were reported in a *POLE* carrier with carcinoma and polyposis [68]. Interestingly, CALMs were reported in one-third of biallelic *POLE* mutation carriers with IMAGe syndrome (intrauterine growth restriction, metaphyseal dysplasia, adrenal hyopoplasia congenita, and genital anomalies; MIM 618336). Thus, CALMs may be related to a mutated *POLE*. It is of note that missense variants in the exonuclease domain of *POLD1* and *POLE* have been associated with PPAP, while there is no evidence that truncating variants are associated with PPAP. Bi-allelic pathogenic loss-of-function (LoF) variants of *POLE* have been detected in IMAGE syndrome (OMIM 614732), which is characterized by intrauterine growth retardation, metaphyseal dysplasia, adrenal hypoplasia congenita, and genital anomalies, as well as in FILS syndromes (OMIM 615139), which is characterized by malar hypoplasia, livedo, short stature, and immunodeficiency.

### 2.9. PTEN Hamartoma Tumor Syndrome

*PTEN* hamartoma tumor syndrome (PHTS), a dominantly inheritable syndrome, is caused by pathogenic germline *PTEN* variants. It is associated with an increased risk of the development of several forms of cancer, including colorectal, breast, thyroid, uterine, and renal cancers. In most cases, the syndrome is clearly recognizable by the presence of several dysmorphological and behavioral manifestations and benign tumors. They include macrocephaly (occipital frontal circumference ≥ 97th percentile); thyroid lesions (e.g., adenoma, multinodular goiter); intellectual disability (IQ ≤ 75) or developmental delay; autism spectrum disorder; Other features include: mucocutaneous lesions (trichilemmomas, cutaneous facial papules, oral mucosal papillomatosis, acral keratoses, and palmoplantar keratoses); ganglioneurinomas, hamartomatous, or other intestinal polyps; Lhermitte–Duclos disease (LDD); fibrocystic disease of the breast; lipomas; fibromas; genitourinary malformation; vascular features (such as arteriovenous malformations or hemangiomas); freckling of the penis; and uterine fibroids [69,70]. As up to 44% of patients have de novo mutations [71] with no family history, these manifestations should prompt physicians to conduct *PTEN* germline mutation analysis so cancer surveillance can be offered in time. This is important, because almost 30% of patients with a germline *PTEN* mutation can be identified using the NCCN criteria for PHTS, but only one-third of patients with PHTS fulfill these criteria [72]. The Cleveland Clinic Risk Assessment Tool, a scoring system available online, has been shown to be more accurate than the NCCN criteria for the diagnosis of PHTS [73]. 

In children, PHTS should be considered in the presence of autism spectrum disorder or developmental delay when it co-occurs with macrocephaly. At age 30–40 years, most patients have mucocutaneous and oral features. A Dutch study assessing the phenotypic characteristics of 81 children and 86 adults with PTHS followed yearly at an expert center over a period of 23 years identified macrocephaly in 100% of the pediatric patients and 67% of the adult patients. The prevalence of multinodular goiter was ∼50% in children and gradually increased to >90% in adults. Similar rates were observed for the oral features, namely, gingival hypertrophy, high palate (adults only), and oral papillomas. A positive score (in adults) for two out of three of these characteristics yielded a sensitivity of 100% (95% CI 94–100%) [74]. The authors suggested that the presence of these features should prompt physicians and dentists to consider a further assessment for PHTS in adults [74]. These findings were supported by an Italian study of 20 patients with PTHS (16 adults and 4 children) wherein all patients had the typical mucocutaneous features of papules on the trunk or extremities (74%), papules in the oral mucosa (68%), facial papules (58%), and acral/palmoplantar keratosis (58%) [75,76]. 

LDD, a rare slow-growing cerebellar dysplastic gangliocytoma, is found in 6–15% of patients with PHTS. It is usually diagnosed in young adulthood, although it can also present in childhood and older age. Germline *PTEN* mutations seem to be more common in unselected LDD tumors from adults than those from children [70]. A review of 14 children with LDD showed that 3 had a clinical diagnosis of PHTS, whereas 8 had no signs of the disease; for 3, data were insufficient for a diagnosis [77]. A study in China in 2019 followed 12 patients of mean age 28.0 ± 14.8 years with LDD, of which 4 of the patients, aged 16–18 years, were diagnosed with PHTS [78]. 

Multiple polyps with a mixed and/or unusual histology should alert gastroenterologists and pathologists to a possible diagnosis of PTHS. Polyps of different histological types are seen in virtually all patients with PTHS, usually throughout the gastrointestinal tract. Polypoid ganglioneuroma is the most specific for PHTS, but hamartomas (also juvenile), serrated (hyperplastic), adenomatous, lipomatous, and inflammatory polyps can be seen as well [70,79]. 

### 2.10. STK11-Associated Peutz–Jeghers Syndrome (PJS) 

Peutz–Jeghers syndrome (PJS) is an autosomal dominant inherited disorder characterized by the presence of hamartomatous polyps anywhere along the gastrointestinal tract, and, in some cases, outside of it. Carriers are also at increased risk mainly of breast, pancreas, and uterine cancer. The original description of this disorder is credited to Peutz in 1921. In 1949, Jeghers noted the association of PJS with pigmentation, polyposis, and an increased risk of invasive malignancy.

Approximately 50% of cases of PJS are caused by a germline mutation in *LKB1*/*STK11* [80,81]. The characteristic mucocutaneous pigmentation in an individual with a first-line family history of PJS is one of the diagnostic criteria for the syndrome. The pigmentation takes the form of florid freckling of one or more of the following sites: lips, buccal mucosa, vulva, fingers, and toes. It develops during the first decade of life, but usually fades from the third decade onwards. Notably, some patients with PJS polyps appear never to have had pigmentation, despite frequent, detailed medical examinations. Conversely, a few patients with pigmentation are never known to have PJS polyps [82].

The lifetime cancer risk of PJS is about 93% [83]. The risk specifically of a sex cord tumor with annular tubules may be associated with the overproduction of progesterone [84].

## 3. The Clues

It is valuable to estimate the prevalence in the general population of benign clinical features associated with cancer predisposition syndromes to help distinguish those suggestive of a diagnosis, such as bilaterality, CHRPE, or multiple osteomas in FAP, and LLD, oral mucosa papules, and acral/palmoplantar keratosis in PHTS, from those that may only support the suspected diagnosis, such as macrocephaly and mental retardation (in PHTS) or oligodontia (in syndromes due to mutated genes associated with *Wnt1* (*APC*, *AXIN2*). (Table 2). The main selected features are shown in Figure 1 and elaborated below.

### 3.1. Congenital Hypertrophy of the Retinal Pigmented Epithelium

CHRPE consists of discrete, flat, pigmented lesions with a depigmented halo of the retina that do not cause clinical manifestations, are not age-dependent (as they are congenital), and may be present at birth (80%) or shortly after birth. The visualization of the lesions may require an examination of the ocular fundus with an indirect ophthalmoscope through a dilated pupil. CHRPE is found in 1.2–4.4% of the general population [34]. Though reported in the majority of mutated *APC* carriers with FAP, CHRPE alone is not pathognomonic for underlying germline *APC* mutations. However, the presence of multiple or bilateral CHRPE lesions may indicate that the at-risk family member has inherited FAP [85]. It has been suggested that the evaluation of patients for CHRPE, which is easily detectable with minimally invasive means, may serve as a useful screening method in overburdened healthcare systems or where genetic testing is unfeasible [35]. CHRPE was also reported in 3/55 (5.5%) patients with MAP, although the authors claimed they were unable to confirm that it was of the type associated with polyposis [65]. 

### 3.2. Osteomas

Osteomas are benign tumors characterized by compact lamellar cortical or cancellous bone. They are found most commonly on the skull and mandible, but may occur in any bone of the body. Osteomas do not usually lead to adverse clinical outcomes and do not become malignant. The vast majority develop sporadically. As most lesions are discovered incidentally, the true incidence is unknown. According to one report, osteomas were detected in 0.42% of 16,000 sinus radiographs and 6.4% of 1724 computed tomography scans in patients with a sinus disease [86]. Sporadic osteomas tend to be solitary. Multiple osteomas of the calvarium and mandible should raise the possibility of FAP, which is associated with multiple osteomas in 20–60% of affected individuals. They may appear in children prior to the development of colonic polyps. 

### 3.3. Dental Abnormalities

The literature shows a potential association between dental anomalies, especially tooth agenesis and neoplastic changes, and CRC predisposition syndromes, highlighting the importance of collaborative care and the awareness of dentists and primary physicians.

Types of dental abnormalities include impacted teeth, congenitally missing teeth, supernumerary teeth, dentigerous odontogenic cysts (associated with the crown of an unerupted tooth), compound odontomas, and hypercementosis. Dental abnormalities have been reported in 30–75% of patients with FAP [87], but precise numbers are unavailable. A solitary lesion may be sporadic. Some authors suggested that the presence of three such lesions or of one or more lesions in association with other extracolonic features should raise a suspicion of Gardner syndrome [87]. 

Supernumerary teeth occur in up to 4% of the general population. A single extra tooth is the most common manifestation; three or more supernumerary teeth are found in only 1% of cases [88]. The prevalence of supernumerary teeth in patients with FAP ranges from 11% to 27% [8]. 

Tooth agenesis is the congenital absence of one or more teeth. It is one of the most common developmental dental anomalies in humans, with a reported incidence of 3–10%, depending on the population studied and higher in females than males. Third molars are the most commonly missing permanent teeth, followed by mandibular second premolars [89]. Syndromic tooth agenesis is associated with ectodermal dysplasia, Down syndrome, Soto syndrome, Van der Woude syndrome and other syndromes, and with various conditions of craniofacial anomalies [90].

Although odontogenesis and tumorigenesis seem to be unrelated processes, studies have shown a higher incidence of cancer in general and of ovarian or breast cancer in particular in families with tooth agenesis or hypodontia. This finding suggests an overlap of genetic determinants with molecular pathways (reviewed by [91]) with the plausibility of using hypodontia in childhood as a screening tool or marker for the risk of neoplasms. The main two CRC syndromes associated with tooth agenesis are FAP and *AXIN2* oligodontia. 

Other oral features may be suggestive of PHTS, such as gingival hypertrophy, high palate (adults only) and oral papilloma. The combination of any two of these was found to yield a sensitivity of 100% for PHTS. In addition, the macular melanin deposits typical of PJS often involve the lips and buccal mucosa. Lesions may also develop on the gingiva, palate, and tongue. Histologically, the oral lesions show an increase in melanin in the basal layer, without an obviously increased melanocyte count. The spots are usually found to fade or disappear after age 70 years [74].

### 3.4. Desmoid Tumors

Desmoid tumors are rare fibroblastic tumors that can develop in any region of the body. Their estimated incidence in the general population is 2–4 per million per year [92]. Although desmoid tumors do not display a metastatic potential, their locally aggressive nature can cause severe outcomes. Most cases appear sporadically, mainly on the trunk. Around 5–15% of desmoid tumors, usually mesenteric and presenting as multiple lesions, are associated with FAP [93]. FAP-associated tumors tend to appear earlier than sporadic desmoid tumors, and their appearance is correlated with the location of the *APC* [29]. Interestingly, in a cohort of patients with mesenteric desmoids, half of those without FAP who underwent next-generation sequencing with multigene panels were found to carry a pathogenic variant in *CHEK2*, *BLM*, *ERCC5*, *MSH6*, or *PALB2* [94].

### 3.5. Skin Lesions

The literature shows some association between various skin lesions and CRC predisposition, especially lenitgines and CALMs, which are shared among a few of the syndromes. PJS is the prototype of these syndromes. 

Lentigines (from the Latin lentigo meaning small lentil) consist of flat-pigmented macules characterized by a small size, irregular borders, and discrete markings in different shades of brown and black. In addition to PJS, lentigenoses appear in PHTS. They have also been identified in such rare conditions as Carny complex, LEOPARD (multiple lentigines, electrocardiographic conduction abnormalities, ocular hypertelorism, pulmonic stenosis, abnormal genitalia, growth retardation, and sensorineural deafness), Noonan syndrome, Watson syndrome, and McCune–Albright syndrome. There is a significant clinical overlap between PJS and Carny complex [95]. In general, lentigines associated with genetic diseases develop at a young age, often increase in number during adolescence, and are not restricted to sun-exposed areas. They should be distinguished from solar lesions which often develop after the third decade of life, increase with advancing age, and as the name implies, are found almost exclusively on sun-exposed areas. Lentigines typically do not darken with sun exposure and may be distributed on distinct anatomic locations [96].

### 3.6. Café au Lait Macules

CALMs are flat, hyperpigmented lesions varying from light brown to dark brown with a diameter ranging from 2 mm to more than 20 cm. They are detected in 2.7% of newborns [97] and 28% of school-age children [95]. Three or more CALMs are observed in about 1% of children [98] and 14% of adults [99]. The presence of one to two lesions in a healthy child is not of any concern, but more than five lesions measuring >5 mm in young children, or >15 mm in older children, should raise a suspicion of an underlying pathology [100].

One group observed multiple CALMs in NF1, which is also associated with such non-tumoral manifestations as skinfold freckling and iris Lisch nodules [101], and in other cancer predisposing syndromes such as legius syndrome, caused by mutated *SPRED1* [102], McCune–Albright syndrome, Bloom syndrome, Fanconi anemia, tuberous sclerosis, and ataxia-telangiectasia [103]. More recently, CALMs were observed in 62–97% of carriers of the biallelic mutation in MMR genes causing CMMRD. There is a phenotypic overlap between CMMRD and NF1. Approximately 20% of patients with CMMRD exhibit more than one feature of NF1, and CMMRD is ultimately diagnosed in 0.41% of pathogenic *NF1*/*SPRED1* variant-negative children suspected of having sporadic NF1 [104]. CALMs may also be related to mutated *POLE*. One report described a 14-year-old boy with polyposis and rectosigmoid carcinoma who also had multiple CALMs and a pilomatricoma mimicking the clinical phenotype of CMMRD. He was found to carry a novel pathogenic *POLE* germline mutation, p. (Val411Leu). The authors suggested that this variant may confer a more severe phenotype than previously reported [105].

CALMs were reported in a high proportion of children with JPS, sporadic juvenile polyps, and FAP. Of the total 14 patients, 8 (57.1%) had at least one spot and 4 (28.6%) had multiple spots, compared to rates of 28.5% and 5.2%, respectively, in the general population. Both these differences were statistically significant (CI 28.9–82.3%, *p* < 0.023 and CI 8.4–58.1%, *p* < 0.005) [106]. The frequency of a solitary CALM was not significantly increased in children with pediatric solid tumors [103]. 

## 4. Pathways

How are nonmalignant features of inherited CRC syndromes mediated by the disease-causing genes? We will discuss five major signaling pathways: Wnt, the mammalian target of rapamycin (mTOR), the transforming growth factor beta (TGF-β), base-excision repair (BER), and DNA repair homologous recombination.

### 4.1. Wnt Signaling Pathway

The nonmalignant features identified in patients with FAP and AFAP, caused by germline mutations in *APC*, and oligondotia–CRC syndrome, caused by alterations in *AXIN2*, point to the involvement of the Wnt signaling pathway, a primordial instructive genetic program that is evolutionarily conserved throughout the animal kingdom [107]. The *Wnt* genes encode a large family of secreted protein growth factors. The first one, *Wnt1*, initially called *Int1*, was identified 40 years ago as an oncogene activated by mouse mammary tumor proviral DNA at the *Wnt1* locus, resulting in mammary tissue hyperplasia and tumor formation [108]. Wnt signaling is essential for coordinating complex cellular behaviors during development, regeneration, and cellular homeostasis, where it mediates cell proliferation, polarity, differentiation, cell motility, and stem cell activity [109]. Given the biological omnipresence of the Wnt pathway, it is not surprising that mutations in its components are frequently involved in different types of cancer. The first link was established when the *APC* gene was discovered as the germline cause of FAP [110]. When both *APC* alleles are inactivated, the Wnt pathway is activated in colonic cells and drives polyp formation. Central to the mediation of Wnt signals is the regulation of β-catenin level and localization. On receipt of a Wnt signal by the cells, β-catenin is stabilized and binds transcription factors regulating the expression of Wnt target genes. In the absence of Wnt signaling, β-catenin is subjected to phosphorylation and subsequent degradation by the action of a multiprotein complex [111] organized by *APC* and *AXIN1* or its homolog *AXIN2*, which serve as part of the complex scaffold. Mutations that facilitate the escape of β-catenin from the action of the degradation complex, such as those affecting *APC* or *AXIN2* [44], lead to cancer due to an increased transcription of Wnt target genes. 

In the eye, Wnt signaling is tightly regulated in multiple developmental processes, including the formation of the lens, retinal pigment epithelium (RPE), ciliary margin, dorsoentral pattern in the optic cup, and retinal vascular system. Wnt signaling regulates RPE adhesion, morphogenesis, and pigmentation [112], which may explain the retinal features in patients with FAP. 

The manifestations of some ciliopathies, such as polycystic kidney disease and Bardet–Biedl syndrome, are similar to the extracolonic manifestations of FAP, suggesting that FAP is a cilia-related disorder [113]. Molecularly, β-catenin is the common downstream target in the Wnt signaling pathway, and the *APC* mutations causing the extracolonic manifestations are located mainly in the area of the protein responsible for binding to β-catenin, causing its overexpression [114]. Polycystic kidney disease and CHRPE are also induced by the same β-catenin overexpression in animal models [115,116] 

### 4.2. mTOR Signaling Pathway

The mTOR signaling pathway is implicated in PJS, due to germline mutations in *STK11*, by the presence of mucocutaneous pigmentation in some patients [117], and in Cowden syndrome, by the presence of glycogenic acanthosis of the esophagus and skin lesions in some patients who carry germline mutations in *PTEN* [118].

The mTOR signaling pathway integrates both intracellular and extracellular signals and serves as a central regulator of cell metabolism, growth, proliferation, and survival; it is deregulated in human diseases such as cancer and type 2 diabetes. The mTOR protein is a serine/threonine kinase that belongs to the phosphoinositide 3-kinase (PI3K)-related kinase family and is conserved throughout evolution [119]. *STK11* is a multifunctional serine/threonine kinase inhibitor of mTOR signaling which plays a role in the regulation of cell cycle progression, cellular energy homeostasis, and cell polarity [120]. *PTEN* is a lipid phosphatase that counteracts PI3K by dephosphorylating phosphatidlinositide-3,4,5-triphosphate. The loss of function of *PTEN* leads to a high level of phosphatidlinositide-3,4,5-triphosphate, which activates signaling targets downstream of PI3K, such as Akt. Given that *PTEN* is inactivated by the Akt-mediated phosphorylation of *TSC2*, its loss would indirectly stimulate the mTOR pathway [121]. However, the link between the dysfunction caused by germline variants of *PTEN* and *STK11* and the clinical manifestations, especially the non-malignant ones, remains largely elusive. 

### 4.3. TGF-β Signaling Pathway

Mucocutaneous telangiectasias is part of the clinical characteristics of *SMAD4*-JPS. The inherited predisposition to JPS is due to germline mutations in *SMAD4* [122] or *BMPR1A* [123].

TGF-β signaling plays a critical role in controlling tissue development, proliferation, differentiation, apoptosis, and homeostasis. It is altered in CRC [124]. The TGF-β superfamily has more than 30 components, divided into the TGF-β-activin-nodal subfamily and the bone morphogenetic protein (BMP) subfamily [125]. *SMAD4* encodes a group of proteins that transduce extracellular signals directly to the nucleus. It is the central mediator of both the TGF-β and BMP signaling pathways. *BMPR1A* is part of the BMP family of transmembrane serine/threonine kinases. Despite its role in bone morphogenetic signaling, *BMPR1A* mutations are for the most part not associated with developmental abnormalities. 

### 4.4. BER Signaling Pathway

The BER signaling pathway is implicated in MAP by the presence of pilomatricomas, and in seborrheic keratosis in patients with *NTLH1*-associated polyposis.

Tumorigenesis can be regarded as an imbalance between the mechanisms of cell cycle control and mutation. The BER pathway is the main mechanism involved in the removal and repair of oxidized DNA bases [126]. OGG1 (8-oxo-dG DNA N-glycosylase 1) removes 8-oxo-dG from the DNA, and mutY DNA glycosylase (*MUTYH*) excises misincorporated adenines opposite to 8-oxo-dG via replicative DNA polymerases α, δ, and ε. Both glycosylases suppress tumorigenesis by preventing mutagenic G:C > T:A transversions. This combination with the specific sequence context (transversions occurring preferentially in AGAA or TGAA motifs) has been assigned a novel mutational signature for MAP tumors, termed *signature 36* [127].

*NTHL1*, a DNA glycosylase, initiates the DNA base excision repair of oxidized ring saturated pyrimidine residues [128]. It acts as the first line of defense against genomic mutations caused by environmental carcinogens.

### 4.5. Homologous Recombination Signaling Pathway

*MCM8* and *MCM9* were recently added to the list of genes causing CRC with an autosomal recessive pattern of inheritance [18,54]. Patients also presented with premature ovarian failure [54] or infertility problems [18]. Indeed, germline mutations in these two genes have also been previously associated with premature ovarian failure [129,130]. DNA interstrand crosslinks (ICL) are highly toxic lesions that arrest the replication fork to initiate the repair process during the S phase of vertebrates. Proteins involved in Fanconi anemia, nucleotide excision repair, and translesion synthesis collaboratively lead to homologous recombination repair. The replicative-helicase-related MCM family of proteins, *MCM8* and *MCM9*, form a complex required for homologous recombination repair induced by ICL [131]. It is plausible that gametogenesis and tumorigenesis share several genetic factors, especially those involved in stem cell renewal/differentiation, mismatch repair mechanisms, and apoptosis.

### 4.6. Mismatch Repair System

Pathogenic variants in four of the MMR genes can cause Lynch syndrome (reviewed in [132]). The DNA MMR system is necessary for the maintenance of genomic stability. The main function of MMR proteins is to maintain genomic stability by correcting single-base mismatches and insertion/deletion loops (IDL) that may arise during replication. The most common cause of MMR-deficiency in human cancers is somatic hypermethylation of the *MLH1* promoter, while others are due to loss of heterozygosity from the germline allele in any one of the MMR genes. The malfunction of MMR results in a mutator phenotype and microsatellite instability (MSI) characteristic of most tumors from Lynch syndrome and some 15% of sporadic tumors. MMR proteins also recognize diverse types of endogenous and exogenous damage and correct the lesions, or, if this is not possible, signal the DNA damage to cell cycle arrest or apoptosis. MMR proteins regulate genetic recombination by correcting mismatches that may occur in recombination during meiosis and by suppressing recombination between homoeologous sequences during mitosis. MMR proteins promote the somatic hypermutation and class switch of antibody genes. The main mismatch-binding factor in humans is hMutSα, consisting of *MSH2* and *MSH6*, which recognizes single-base mispairs and IDLs. Upon mismatch binding, the hMutS complex undergoes an ATP-driven conformational change into a sliding clamp, and a hMutL heterodimer is recruited. The main hMutL complex is hMutLα, consisting of *MLH1* and *PMS2* and participating in the repair of single-base mismatches and IDLs. When the hMutS–hMutL complex encounters a strand discontinuity, an excision machinery is recruited, the mismatch containing fragment is degraded, and a new strand synthesized. The lifetime risks of cancer are significantly higher in *MSH2* and *MLH1* mutation carriers compared to carriers of *MSH6* or *PMS2* mutations. Tumors with MMRd, due to somatic or germline inactivation, have a unique signature (SBS6 v3) [133].

## 5. Conclusions

This review describes nonmalignant features associated with inherited CRC syndromes that may serve to facilitate their diagnosis. By improving their understanding of the spectrum of clinical features and their underlying processes, clinicians can gain insight into the pathogenesis of the various CRC syndromes and increase their awareness of additional underdiagnosed manifestations that may occur in carriers. Better understanding of the genetic background of these extracolonic features, as well as the biological relevance of the signaling pathways behind them, may promote treatment options not only for cancer prevention but also for the other clinical features that may pose a significant morbidity and clinical burden, such as *SMAD4*-associated HHT and *APC*-associated desmoid tumors. The treatment of these manifestations can also significantly improve the quality of life of carriers.

## Figures and Tables

**Figure 1 cancers-14-00628-f001:**
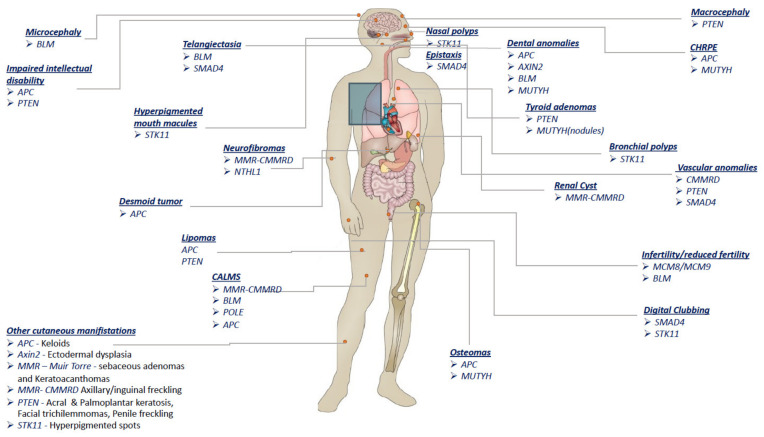
Graphical image of the various benign features associated with the predisposition colon cancer syndromes.

**Table 1 cancers-14-00628-t001:** Ways in which awareness of nonmalignant features of CRC predisposition syndromes are important for clinicians.

Clinical diagnosis	Raises clinical suspicions Influences decision to refer patients for genetic testing Makes early intervention possible when necessary. Prompts surveillance of asymptomatic at-risk patients.
Test choice	Single gene or multi-gene panel CNV testing Search for mosaicism
Test interpretation	VUS or likely pathogenic Mosaicism Family segregation
Variant class interpretation	*POLE*-Missense (PPAP) *SMAD4*–LOF (JPS-HHT)

Abbreviations: CRC, colorectal cancer; CNV, copy number variation; VUS, variant of unknown significance.

**Table 2 cancers-14-00628-t002:** Inherited colorectal cancer predisposition syndromes and major non-malignant features.

*Gene*	Syndrome	Inheritance	Non-Malignant Features
*APC*	FAP	AD Inherited (75–80%) De novo (15–20%)	Eyes–CHRPE (>90%) often multiple bilateral Teeth—Dental anomalies: supernumerary teeth, unerupted teeth, dental caries, or odontomas (17%) Skull—osteomas, especially involving the mandibular angle Limbs—endosteal and exosteal osteomas (20%) Skin—Epidermoid inclusion cysts; fibromas; lipomas; lipofibromas, keloids; desmoid tumors (10–25%) Endocrine system—adrenal masses (7–13%) Central nervous system—mildly impaired intellectual abilities.
	AFAP	AD	Eyes—CHRPE (rare) Skin—desmoid tumors (rare)
	5q-	AD Mostly De novo	Central nervous system—mental retardation Face—dysmorphic features Reproductive system—asthenozoospermia (absence of TSSK1B)
*AXIN2*	ODCRCS	AD	Teeth—tooth agenesis (oligodontia) Skin—ectodermal dysplasia; sparse hair and eyebrows
*BMPR1A*	JPS	AD Inherited (33%) De novo (67%)	Not reported
*BLM*	BLOOM	AR	Growth retardation and growth failure Head—microcephaly Face—malar hypoplasia; prominent ears; prominent nose Teeth—absent upper lateral incisors Reproductive system—azoospermia; reduced fertility in females Skin—facial butterfly telangiectasia; spotty hypopigmentation; CALMs; photosensitivity Immune system—immune abnormalities
*GREM1*	HMPS	AD	Not reported
*MCM8* *MCM9*		AR	Reproductive system—primary ovarian insufficiency/azoospermia
*MMR genes* *MLH1, MSH2, MSH6, PMS2*	LYNCH	AD	Skin—sebaceous adenomas and keratoacanthomas (Muir–Torre)
*MMR genes*	CMMRD	AR	Skin—CALMs (62–97%); neurofibromas; axillary/inguinal freckling (10%) Central nervous system—agenesis of the corpus callosum; gray matter heterotopia; intracerebral cyst; interhemispheric cyst Vascular system—developmental vascular abnormalities Kidneys—renal cysts Autoimmune system—pediatric SLE
*MSH3*	FAP4	AR	Not reported
*MUTYH*	MAP	AR	Endocrine system—thyroid nodules; benign adrenal lesions (18%) Eyes—CHRPE (5.5%) Limbs—osteomas Teeth—dental anomalies; jaw bone cysts
*NTHL1*	NAP	AR	Skin—hemangiomas; neurofibromas Genitourinary system—ovarian cysts Liver—liver cysts
*POLE*	PPAP	AD	Skin—CALMs;
*POLD1*	PPAP	AD	Not reported
*PTEN*	PHTS	AD De novo (44%)	Head—macrocephaly Mouth—oral papillomas Vascular system—vascular anomalies (50% of patients); intracranial developmental venous and arteriovenous malformations Genitourinary system—genitourinary malformation; uterine fibroids Skin—multiple facial papules; acral keratosis; palmoplantar keratosis; facial trichilemmomas; lipomas; fibromas; penile freckling Central nervous system—Cerebellar gangliocytoma manifesting as seizure and tremor (Lhermitte-Duclos disease; 6–15%); mental retardation (12%); autism Endocrine system—goiter; thyroid adenomas
*RNF43*	SSPCS	AD	Not reported
*SMAD4*	HHT	AD	Vascular system—AVMs (76%) Hands—digital clubbing (50%) Skin—telangiectasia (57%) Nose—epistaxis (61–71%)
*STK11*	PJS	AD	Nose—nasal polyps Mouth—hyperpigmented macules of lips and buccal mucosa (65%) Respiratory system—bronchial polyps Genitourinary system—ovarian cysts; gynecomastia with sertoli cell tumors Hands—clubbing of fingers Skin—dark blue to dark brown melanocytic macules (which fade with age); hyperpigmented spots on hands/digits (especially palms; 73%), arms, feet (especially plantar areas), legs, or lips
*TP53*	LI-FRAUMENI	AD	Not reported

Numbers are taken from OMIM and GeneReviews. Abbreviations: AFAP—attenuated FAP; AVMs—arteriovenous malformations; CALMs—café-au-lait macules; CHRPE—congenital hypertrophy of the retinal pigment epithelium; CMMRD—constitutional mismatch repair deficiency syndrome; FAP—familial adenomatous polyposis; HHT—hereditary hemorrhagic telangiectasia; HMPS—hereditary mixed polyposis syndrome; JPS—juvenile polyposis; MAP—*MUTYH*-associated polyposis; NAP—*NTHL1*-associated polyposis; ODCRCS—oligodontia–CRC syndrome; PPAP—polymerase proofreading-associated polyposis; PHTS—*PTEN* hamartoma tumor syndrome; PJS—Peutz–Jeghers syndrome, SLE—systemic lupus erythematosus; SSPCS—sessile serrated polyposis cancer syndrome.

## Data Availability

Not applicable.

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
