# Peer review of "Nonmalignant Features Associated with Inherited Colorectal Cancer Syndromes-Clues for Diagnosis"

_cancers, 2022, doi:10.3390/cancers14030628_

Round 1
Reviewer 1 Report
In this manuscript, the authors review the non-malignant comorbidities associated with CRC predisposition syndromes. I think there are nice summaries of these features in Table 2 and Fig 1 and discussions in the text which will be helpful to clinicians diagnosing these syndromes.
Table1:
- variants of unknown significance are most commonly abbreviated as VUS in the literature (not VOUS).
Table2:
- What was the criteria for inclusion of a gene in table 2? Some genes are well established while others seem to have much less evidence (i.e., GALNT12, MCMs, etc). GALNT12 is not mentioned anywhere else except for that one line in the table.
The most recent 2021 ACMG guidelines for inherited CRC (Genetics in Medicine, Jun 2021) includes some other important genes, including MSH3 and ATM. Please reconcile your list with theirs in either the main result or in the discussion.
- Why do some clinical characteristics have percentages annotated, while others don't?
- Some of the rare comorbidities are listed in Table 2 (like for AFAP), while some aren't (like for SMAD4-associated HHT).
- BMR1A should be BMPR1A
- Please include the MMR gene names in Table 2, if not in the table itself, at least in the table legend/abbreviations.
- To help clinicians with choosing genetic tests and interpretation of test results, it would be helpful to include the types of mutation that are known to be pathogenic (loss of function, activating, deletions, etc) in Table 1, and if any one test is most sensitive or specific for that syndrome. This would also be helpful for example in the section 2.8 POLE discussion because germline POLE missense proofreading mutations seem different from the POLE deficiency-associated mutation seen IMAGe syndrome.
In the text, Increased cancer risk is discussed for some genes (PTEN) but not others (MMR genes, POLE that have increased risk for endometrial and ovarian cancers, etc). Please consistently address this across the syndromes.
Fig 1 - please color code the gene names? Then it will be easier to see when a gene is noted multiple times in the figure.
line 293 - anoma;ies typo
Author Response
In this manuscript, the authors review the non-malignant comorbidities associated with CRC predisposition syndromes. I think there are nice summaries of these features in Table 2 and Fig 1 and discussions in the text which will be helpful to clinicians diagnosing these syndromes.
Table1:
- variants of unknown significance are most commonly abbreviated as VUS in the literature (not VOUS).
Thank you - corrected
Table2:
- What was the criteria for inclusion of a gene in table 2? Some genes are well established while others seem to have much less evidence (i.e., GALNT12, MCMs, etc). GALNT12 is not mentioned anywhere else except for that one line in the table.
The most recent 2021 ACMG guidelines for inherited CRC (Genetics in Medicine, Jun 2021) includes some other important genes, including MSH3 and ATM. Please reconcile your list with theirs in either the main result or in the discussion.
Thanks for your important comment and referral. Gene choosing was based on NCCN updated guidelines, ACMG guidelines for inherited CRC, recommended genes in the different national and commercial panels, a comprehensive reviews and personal knowledge. When in debate, we chose to include genes with extracolonic non-malignant features that to which awareness can contribute to diagnosis (such as infertility in MCM9)
Following your suggestion we have deleted GALNT12. We agree that current knowledge is not sufficient to include it in the table. We added MSH3 to the table, however, as only two families were studied to depth till now, we did not elaborate in the text. With regard to ATM, while there is plenty evidence of it, being a susceptibility gene for breast and pancreas cancer, we do not feel there is enough convincing evidence, nor surveillance guidelines for CRC early detection, though the gene is known and studied for a long time.
- Why do some clinical characteristics have percentages annotated, while others don't?
We have provided percentage if appears in the literature, however, for many of the clinical features there is no information about percentage .
- Some of the rare comorbidities are listed in Table 2 (like for AFAP), while some aren't (like for SMAD4-associated HHT).
SMAD4 HHT is elaborate in table 2 . Please see the line before STK11
- BMR1A should be BMPR1A
Corrected
- Please include the MMR gene names in Table 2, if not in the table itself, at least in the table legend/abbreviations.
Added
- To help clinicians with choosing genetic tests and interpretation of test results, it would be helpful to include the types of mutation that are known to be pathogenic (loss of function, activating, deletions, etc) in Table 1, and if any one test is most sensitive or specific for that syndrome. This would also be helpful for example in the section 2.8 POLE discussion because germline POLE missense proofreading mutations seem different from the POLE deficiency-associated mutation seen IMAGe syndrome.
Thank you for this important remark – we have added our view in the table and in the text.
In the text, Increased cancer risk is discussed for some genes (PTEN) but not others (MMR genes, POLE that have increased risk for endometrial and ovarian cancers, etc). Please consistently address this across the syndromes.
Thank you for this comment – we have indeed added a very brief description when relevant to the other conditions as well, for consistency
Fig 1 - please color code the gene names? Then it will be easier to see when a gene is noted multiple times in the figure.
Thank you. We colored the main AD genes that appear more than once
line 293 - anoma;ies typo
Corrected

Reviewer 2 Report
The revised manuscript is a comprehensive and up-to-date review of hereditary colorectal cancer syndromes. The review focuses mainly on the clinical manifestations associated with the different syndromes but also reviews the pathways involved in them. In this sense, we fault a subsection on MMR in section 4.
We consider that the manuscript contains useful information for researchers in the field.
Supplementary comments: being a review, there are no experiments that can be refuted or completed. I consider it to be an excellent review, very up-to-date and therefore useful to researchers in the field. The review is balanced and the information provided is up-to-date. The objective is to show that the presence of some clinical manifestations should lead to suspicion of the presence of predisposing syndromes to colorectal cancer.
I believe that the manuscript meets this objective.
Section 4 is the weakest, providing very summary information. For this reason, I consider that, if it is maintained, the MMR pathway should be included.
Author Response
Reviewer 2
The revised manuscript is a comprehensive and up-to-date review of hereditary colorectal cancer syndromes. The review focuses mainly on the clinical manifestations associated with the different syndromes but also reviews the pathways involved in them. In this sense, we fault a subsection on MMR in section 4.
We consider that the manuscript contains useful information for researchers in the field.
Supplementary comments: being a review, there are no experiments that can be refuted or completed. I consider it to be an excellent review, very up-to-date and therefore useful to researchers in the field. The review is balanced and the information provided is up-to-date. The objective is to show that the presence of some clinical manifestations should lead to suspicion of the presence of predisposing syndromes to colorectal cancer.
I believe that the manuscript meets this objective.
Section 4 is the weakest, providing very summary information. For this reason, I consider that, if it is maintained, the MMR pathway should be included.
Thank you for this comment. A concise, though relevant summary was added.

Reviewer 3 Report
This is an interesting review where the authors described the nonmalignant characteristics that may impact the colorectal cancer predisposition syndromes. Basically, they have mentioned the extracolonic features such as dermal, dental, reproductive etc. that are closely linked to this cancer. Overall, the presentation of the information in regard to the topic was nice and well organized.
Author Response
no comments for responses